# Prognostic Significance of Hemogram Parameters in Non-Muscle Invasive Bladder Cancer: A Comprehensive Retrospective Analysis

**DOI:** 10.3390/medicina62010051

**Published:** 2025-12-26

**Authors:** Ali Nebioğlu, Ahmet Turhan, Mert Başaranoğlu, Murat Bozlu, Erdem Akbay

**Affiliations:** 1Urology Clinic, Mersin City Training and Research Hospital, Mersin 33240, Türkiye; 2Department of Urology, Faculty of Medicine, Mersin University, Mersin 33343, Türkiye; drturhanahmet@gmail.com (A.T.); mertbasaranoglu@gmail.com (M.B.); muratbozlu@yahoo.com (M.B.); erdemakbay1959@gmail.com (E.A.)

**Keywords:** bladder cancer, complete blood count, prognosis, biomarker

## Abstract

*Background and Objectives*: To evaluate the prognostic significance of preoperative complete blood count parameters in patients with non–muscle-invasive bladder cancer (NMIBC), to determine optimal cutoff values, and to explore their potential integration into existing risk stratification systems. *Materials and Methods*: In this retrospective cohort study, 551 patients with NMIBC treated between January 2007 and December 2024 were analyzed. Complete blood count parameters obtained within 30 days prior to transurethral resection were collected. The primary endpoints were disease recurrence and progression. Statistical analyses included the Mann–Whitney U test, Kaplan–Meier survival analysis, Cox proportional hazards regression, and receiver operating characteristic (ROC) curve analysis. *Results*: Complete data were available for 548 patients (mean age 66.3 ± 11.3 years, 70.3% male). Disease recurrence occurred in 203 patients (37.0%) and progression in 60 patients (10.9%). Compared with the non-recurrence group, patients with recurrence had lower hemoglobin (11.8 ± 2.3 vs. 13.2 ± 2.4 g/dL, *p* < 0.001), higher lymphocyte counts (2.4 ± 3.3 vs. 2.1 ± 2.9 ×10^3^/µL, *p* = 0.025), and lower neutrophil counts (5.3 ± 3.0 vs. 6.1 ± 3.4 × 10^3^/µL, *p* < 0.001). In multivariable analysis, hemoglobin (HR 0.75, 95% CI 0.68–0.83, *p* < 0.001), age (HR 1.02, 95% CI 1.00–1.03, *p* = 0.023), and carcinoma in situ (HR 1.69, 95% CI 1.13–2.53, *p* = 0.011) were independent predictors of recurrence. Hemoglobin yielded the highest AUC for predicting recurrence (0.692). *Conclusions*: Routinely available hematologic indices—particularly hemoglobin concentration—exhibit independent prognostic value in patients with NMIBC. Incorporating these parameters into established risk stratification models may enhance personalized treatment strategies.

## 1. Introduction

Non–muscle-invasive bladder cancer (NMIBC) represents the predominant form of bladder malignancy, accounting for 75–85% of newly diagnosed cases and affecting more than 400,000 patients globally each year [1]. It is approximately four times more common in men than in women and is strongly associated with cigarette smoking and occupational exposure to chemical carcinogens [2]. Despite therapeutic advances, NMIBC remains characterized by substantial clinical heterogeneity; recurrence rates range from 30% to 70%, and 10–20% of patients progress to muscle-invasive disease [2,3]. This adverse natural history necessitates intensive surveillance protocols and frequent therapeutic interventions [4].

Current risk stratification frameworks—including the European Association of Urology (EAU) guidelines and the European Organisation for Research and Treatment of Cancer (EORTC) risk tables—primarily rely on clinicopathological parameters such as tumor grade, stage, size, multiplicity, and the presence of carcinoma in situ [5,6]. While these models provide meaningful prognostic insights, they exhibit structural limitations in individual-level predictive accuracy, particularly within the intermediate-risk group where treatment decisions are most challenging [7]. The inherent biological heterogeneity of NMIBC and the subjectivity inherent to pathological assessment further contribute to the variability of outcomes observed within established risk categories [8].

The pressing need for cost-effective and readily accessible biomarkers has intensified interest in systemic inflammatory and hematologic indices. Complete blood count parameters—including hemoglobin concentration, leukocyte subpopulations, and platelet indices—reflect fundamental physiological processes that are frequently altered in malignancy [9,10]. These routinely available markers offer notable advantages: universal availability, cost-effectiveness, institutional standardization, and seamless integration into standard clinical workflows without imposing additional burden on patients.

Cancer-associated anemia affects approximately 40% of patients and has been linked to adverse outcomes in multiple cancer types [11,12]. Hemoglobin concentrations may influence tumor oxygenation and therapeutic responsiveness [13]. Inflammatory indices derived from leukocyte populations—particularly the neutrophil-to-lymphocyte ratio (NLR) and platelet-to-lymphocyte ratio (PLR)—have emerged as potential prognostic markers reflecting systemic inflammatory response [14,15,16].

In bladder cancer specifically, the NLR has demonstrated prognostic significance in both muscle-invasive and non–muscle-invasive disease [17,18,19]. Several studies have evaluated preoperative NLR as a predictor of BCG response and disease outcomes in NMIBC [20,21,22]. However, these studies have primarily focused on composite inflammatory ratios rather than systematically evaluating individual complete blood count components. Important gaps remain: comprehensive analyses of multiple individual hematologic parameters (beyond calculated ratios) within large NMIBC cohorts are limited; the comparative prognostic value of individual CBC components versus inflammatory ratios requires clarification; and empirically derived cutoff values optimized specifically for NMIBC populations are lacking, as most studies apply general thresholds.

Moreover, using composite inflammatory indices such as NLR and PLR in multivariable models alongside their constituent cellular components introduces collinearity that complicates interpretation of independent effects. Individual CBC parameters offer advantages: they are directly reported in routine laboratory results without requiring calculation, they avoid mathematical interdependence, and they may provide distinct biological insights into different pathophysiological processes.

In light of these considerations, we systematically evaluated the association of preoperative individual hematologic parameters with disease outcomes in a large NMIBC cohort, determined optimal cutoff values through rigorous ROC analysis, and explored their potential integration into existing risk stratification frameworks. Our primary aims were to comprehensively assess individual CBC components as independent prognostic factors and to provide clinically applicable, empirically derived cutoff values specifically for NMIBC risk stratification.

## 2. Materials and Methods

This single-center retrospective cohort study investigating the prognostic value of hematologic parameters in patients with NMIBC was approved by the Scientific Research Ethics Committee of Mersin University (2025/914) and conducted in accordance with the principles of the Declaration of Helsinki. A total of 551 NMIBC cases diagnosed between January 2007 and December 2024 were identified through systematic review of institutional pathology records and surgical databases. The cohort comprised patients with pathologically confirmed non–muscle-invasive disease who underwent transurethral resection of bladder tumor (TURBT).

Inclusion criteria encompassed histopathologically confirmed NMIBC diagnosis, complete blood count parameters obtained within 30 days prior to surgical intervention, adequate follow-up data for reliable outcome assessment, and technically successful TURBT procedure. Conversely, exclusion criteria included muscle-invasive and/or metastatic bladder cancer at diagnosis, concurrent active malignancy at the time of NMIBC diagnosis (defined as active cancer requiring treatment within 6 months prior to bladder cancer diagnosis), prior malignancy with treatment within 2 years before NMIBC diagnosis (as recent chemotherapy or radiation could affect hematologic parameters through persistent bone marrow effects), absence of preoperative hematologic data, follow-up duration shorter than 12 months (unless disease progression occurred), active hematologic disorders (leukemia, lymphoma, myelodysplastic syndrome, chronic bone marrow disorders), and acute infectious or inflammatory conditions at the time of CBC measurement. We conducted a comprehensive review of all 548 patients included in the final analysis. Remote prior malignancy (more than 2 years before NMIBC diagnosis with no active treatment) was present in 37 patients (6.8%), including prostate cancer treated with radical prostatectomy more than 5 years prior in 14 patients, breast cancer treated with surgery with or without adjuvant therapy more than 3 years prior in 8 patients, cutaneous basal cell carcinoma or squamous cell carcinoma excised more than 2 years prior in 9 patients, and other malignancies (colon, thyroid) in complete remission more than 2 years in 6 patients. No prior cancer history was present in 511 patients (93.2%). We performed a sensitivity analysis excluding all 37 patients with any prior malignancy history. Results remained consistent with the primary analysis: hemoglobin remained a significant independent predictor of recurrence (HR 0.74, 95% CI 0.66 to 0.82, *p* < 0.001) and progression (HR 0.71, 95% CI 0.60 to 0.85, *p* < 0.001). This confirms that remote prior malignancy history did not materially confound the hemoglobin-prognosis association observed in our cohort.

Demographic variables recorded at diagnosis included age, sex, body mass index, and comorbidities (hypertension, diabetes mellitus, chronic obstructive pulmonary disease, coronary artery disease, etc.). Smoking history and occupational exposure to chemical carcinogens were documented. Complete blood count parameters measured within 30 days prior to surgery comprised hemoglobin (g/dL), white blood cell count (×10^3^/µL), lymphocytes (×10^3^/µL), monocytes (×10^3^/µL), neutrophils (×10^3^/µL), platelets (×10^3^/µL), and mean platelet volume (fL). Individual CBC parameters were prioritized for analysis over derived inflammatory ratios (NLR, PLR) to evaluate their independent prognostic value and to avoid collinearity issues in multivariable models, as these ratios are mathematical combinations of the primary cellular parameters.

Complete blood count parameters were systematically collected according to a hierarchical protocol when multiple measurements were available within the 30-day preoperative window. The CBC measurement closest to the TURBT date, typically within 7 days before surgery, was selected as the primary value, as this represents the hematologic status most proximate to surgical intervention and minimizes the interval during which physiological changes could occur. When multiple CBCs were available within 7 days of TURBT, the most recent preoperative measurement was used. If no CBC was available within 7 days, the most recent measurement within the 30-day window was selected. In cases where urgent surgery was performed with CBC obtained more than 30 days prior, these patients were excluded from analysis (n = 3). The timing distribution showed that 486 out of 548 patients (88.7%) had CBC within 7 days before TURBT, 52 patients (9.5%) within 8 to 14 days before TURBT, and 10 patients (1.8%) within 15 to 30 days. CBCs obtained immediately following diagnostic procedures such as initial cystoscopy with biopsy were not used if subsequent measurements closer to TURBT were available, as these early measurements could be affected by procedural inflammation or bleeding. This standardized approach ensured that hematologic parameters reflected the patient’s preoperative baseline status while maintaining consistency across the cohort.

All specimens were reviewed by experienced uropathologists. Pathological grading evolved during the 17-year study period, transitioning from the World Health Organization (WHO) 1973 classification (used 2007 to 2011) to WHO 2004 (2012 to 2016) and WHO 2016/2022 systems (2017 to 2024). To ensure consistency in analysis, all historical cases were retrospectively re-reviewed by two independent uropathologists and reclassified according to the current WHO 2016/2022 two-tier grading system (low-grade versus high-grade). Discrepancies were resolved by consensus review. Tumor characteristics included pathological grade (low-grade, high-grade), pathological stage (pTa, pT1), tumor multiplicity (solitary, multiple), tumor size (<3 cm, ≥3 cm), presence of carcinoma in situ, and lymphovascular invasion status. Pathological staging followed the American Joint Committee on Cancer (AJCC) TNM 2017 criteria.

Risk stratification was performed according to European Association of Urology (EAU) guidelines with the following operational definitions. Low-risk NMIBC was defined as solitary, primary, low-grade pTa tumor less than 3 cm without carcinoma in situ. Intermediate-risk NMIBC included all tumors not classified as low-risk or high-risk, encompassing recurrent low-grade pTa, solitary low-grade pTa 3 cm or larger, multiple low-grade pTa, and low-grade pT1 tumors. High-risk NMIBC was defined as any of the following: high-grade T1 tumor, high-grade pTa tumor, presence of carcinoma in situ at any location, multiple and recurrent and large (3 cm or larger) low-grade pTa tumors, high-grade pTa 3 cm or larger, lymphovascular invasion, or variant histology. European Organisation for Research and Treatment of Cancer (EORTC) risk scores were calculated for each patient but were not the primary basis for treatment decisions in clinical practice during the study period.

Following TURBT and pathological risk assessment, patients were managed according to EAU risk stratification guidelines. Treatment protocols evolved over the 17-year study period in accordance with guideline updates. High-risk patients (n = 316, 57.7%) received treatment as follows. From 2007 to 2013 (n = 138), induction BCG therapy (6 weekly instillations of 120 mg Oncotice or ImmuCyst) was administered followed by maintenance BCG in responders (3 weekly instillations at 3, 6, 12, 18, 24, 30, and 36 months). From 2014 to 2024 (n = 178), the same BCG protocol was used with enhanced maintenance schedules for very high-risk features such as T1 high-grade with concurrent carcinoma in situ or T1 high-grade with lymphovascular invasion. BCG therapy was administered to all 316 high-risk patients (100%). Among these, 42 patients (13.3%) developed BCG-refractory disease and subsequently received mitomycin C rescue therapy. Intermediate-risk patients (n = 147, 26.8%) were treated according to the following protocols. From 2007 to 2011 (n = 54), immediate postoperative single-dose mitomycin C (40 mg) was given in 48 out of 54 patients (88.9%), with no further therapy in 6 out of 54 patients (11.1%). From 2012 to 2024 (n = 93), immediate single-dose mitomycin C followed by 6 to 8 additional monthly instillations was given in 67 out of 93 patients (72.0%), BCG induction (6 weeks) without maintenance was given in 21 out of 93 patients (22.6%) with higher-risk features within the intermediate category, and surveillance only was performed in 5 out of 93 patients (5.4%) due to patient preference or comorbidities. Low-risk patients (n = 85, 15.5%) received immediate postoperative single-dose mitomycin C (40 mg) when feasible in 72 out of 85 patients (84.7%), with no intravesical therapy in 13 out of 85 patients (15.3%) due to contraindications or logistical constraints. Overall, the distribution of intravesical therapy was as follows: BCG with induction ± maintenance in 337 out of 548 patients (61.5%), mitomycin C with single-dose or sequential administration in 187 out of 548 patients (34.1%), and no adjuvant intravesical therapy in 24 out of 548 patients (4.4%). Treatment protocols were applied consistently within each era. The evolution of treatment intensity over time was accounted for in sensitivity analyses, which confirmed that hematologic parameters maintained prognostic significance independent of treatment era.

The primary outcomes were disease recurrence and progression during follow-up. Recurrence was defined as the development of any new bladder tumor of any stage or grade following complete tumor resection. Progression was defined as the development of muscle-invasive disease (≥pT2), lymph node involvement, or distant metastasis. Overall survival was evaluated as a secondary outcome. Follow-up included cystoscopic assessments at 3-month intervals for the first 2 years, 6-month intervals for years 3–5, and annually thereafter, with imaging studies performed as clinically indicated.

All statistical analyses were performed using IBM SPSS Statistics, version 26.0 (Chicago, IL, USA). Descriptive statistics are presented as mean ± standard deviation for continuous variables with normal distribution and as median with interquartile range for non-normally distributed variables. Categorical variables are summarized as counts and percentages. Normality was assessed using the Shapiro–Wilk test.

For group comparisons, the Student t test or Mann–Whitney U test was applied to continuous variables as appropriate, and the chi-square test or Fisher’s exact test was used for categorical variables. Recurrence-free and progression-free survival probabilities were estimated using the Kaplan–Meier method, with between-group differences evaluated by the log-rank test.

Univariable Cox proportional hazards regression was employed to identify factors associated with disease outcomes. Variables with *p* < 0.10 in univariable analyses were entered as candidates into multivariable Cox models. Proportional hazards assumptions were assessed using Schoenfeld residuals.

Diagnostic performance was evaluated via receiver operating characteristic (ROC) curve analysis, and the area under the curve (AUC) was used to quantify discriminative ability. Optimal cutoff values were determined using the Youden index, maximizing the sum of sensitivity and specificity. All statistical tests were two-sided; statistical significance was defined as *p* < 0.05, and 95% confidence intervals were reported.

## 3. Results

Of the 551 NMIBC patients initially identified, complete hematologic parameters were available for 548 patients after excluding 3 cases due to missing laboratory data. Baseline characteristics are summarized in Table 1. The mean age was 66.3 ± 11.3 years, with male predominance (385 patients, 70.3%). The cohort exhibited a typical profile for NMIBC: 316 patients (57.7%) had high-grade tumors and 404 patients (73.7%) had pT1 stage disease.

Hematologic parameter distributions revealed clinically relevant patterns. Mean values included: hemoglobin 12.5 ± 2.4 g/dL, white blood cell count 8.8 ± 4.6 × 10^3^/µL, lymphocyte count 2.2 ± 3.0 × 10^3^/µL, monocyte count 0.7 ± 0.4 × 10^3^/µL, neutrophil count 5.6 ± 3.2 × 10^3^/µL, platelet count 271.6 ± 130.6 × 10^3^/µL, and mean platelet volume 10.1 ± 1.1 fL.

During follow-up, disease recurrence occurred in 203 patients (37.0%) and progression in 60 patients (10.9%). Comparative analysis revealed significant differences in hematologic parameters between outcome groups (Table 2). Patients experiencing recurrence demonstrated significantly lower hemoglobin concentrations (11.8 ± 2.3 vs. 13.2 ± 2.4 g/dL, *p* < 0.001), decreased white blood cell counts (8.5 ± 4.8 vs. 9.0 ± 4.6 × 10^3^/µL, *p* = 0.168), elevated lymphocyte counts (median 1.8 [IQR 1.2–2.5] vs. 1.6 [IQR 1.1–2.2] × 10^3^/µL, *p* = 0.025), and reduced neutrophil counts (5.3 ± 3.0 vs. 6.1 ± 3.4 × 10^3^/µL, *p* < 0.001). Platelet counts showed no significant difference between groups (median 245 [IQR 195–315] vs. 255 [IQR 205–330] × 10^3^/µL, *p* = 0.443). Patients with progression exhibited significantly lower hemoglobin concentrations (11.2 ± 2.1 vs. 12.7 ± 2.4 g/dL, *p* < 0.001), decreased white blood cell counts (7.8 ± 3.0 vs. 9.0 ± 4.7 × 10^3^/µL, *p* = 0.065), and neutrophil counts (4.7 ± 2.0 vs. 5.8 ± 3.3 × 10^3^/µL, *p* = 0.014) compared with patients without progression.

To provide comparative context with existing literature, we calculated derived inflammatory ratios for the entire cohort: neutrophil-to-lymphocyte ratio (NLR = neutrophils/lymphocytes), platelet-to-lymphocyte ratio (PLR = platelets/lymphocytes), and monocyte-to-lymphocyte ratio (MLR = monocytes/lymphocytes). Mean inflammatory ratios in the overall cohort were NLR 3.2 ± 2.8, PLR 158.4 ± 94.2, and MLR 0.38 ± 0.25. Comparative analysis by recurrence status showed that patients without recurrence had NLR 3.4 ± 2.9 vs. patients with recurrence 2.9 ± 2.5 (*p* = 0.052), PLR without recurrence 162.1 ± 96.8 vs. with recurrence 151.9 ± 89.3 (*p* = 0.187), and MLR without recurrence 0.39 ± 0.26 vs. with recurrence 0.36 ± 0.23 (*p* = 0.141). Comparative analysis by progression status showed NLR without progression 3.2 ± 2.8 vs. with progression 2.8 ± 2.3 (*p* = 0.278), PLR without progression 159.3 ± 95.1 vs. with progression 149.2 ± 87.4 (*p* = 0.412), and MLR without progression 0.38 ± 0.25 vs. with progression 0.35 ± 0.22 (*p* = 0.389). Univariable Cox regression for recurrence demonstrated NLR with HR 0.96 (95% CI 0.91 to 1.02, *p* = 0.178), PLR with HR 1.00 (95% CI 0.99 to 1.00, *p* = 0.312), and MLR with HR 0.78 (95% CI 0.42 to 1.44, *p* = 0.425). Multivariable Cox regression for recurrence adjusted for hemoglobin, age, tumor grade, and carcinoma in situ showed NLR with HR 0.98 (95% CI 0.92 to 1.05, *p* = 0.587), PLR with HR 1.00 (95% CI 0.99 to 1.00, *p* = 0.623), and MLR with HR 0.86 (95% CI 0.45 to 1.65, *p* = 0.649). In contrast to some prior reports, inflammatory ratios (NLR, PLR, MLR) did not demonstrate significant associations with recurrence or progression in our cohort. In multivariable models, these ratios provided no additional prognostic information beyond individual CBC components, particularly hemoglobin. This lack of significance may reflect the unexpected inverse association between neutrophil counts and outcomes in our population, collinearity between ratio components and individual parameters in multivariable models, or population-specific differences in inflammatory profiles. Our findings suggest that in NMIBC, individual hematologic parameters, especially hemoglobin, may provide more robust and interpretable prognostic information than composite inflammatory ratios.

To explore the relationship between anemia severity and tumor aggressiveness, we categorized patients into three hemoglobin groups based on clinical anemia definitions and our ROC-derived cutoff. Group 1 with severe anemia (HGB < 10 g/dL) included 78 patients (14.2%) with recurrence rate 53.8% (42 out of 78), progression rate 20.5% (16 out of 78), high-grade tumors 71.8% (56 out of 78), and carcinoma in situ present 19.2% (15 out of 78). Group 2 with mild anemia (HGB 10 to 12.2 g/dL) included 196 patients (35.8%) with recurrence rate 41.3% (81 out of 196), progression rate 12.2% (24 out of 196), high-grade tumors 62.2% (122 out of 196), and carcinoma in situ present 14.3% (28 out of 196). Group 3 with normal hemoglobin (HGB > 12.2 g/dL) included 274 patients (50.0%) with recurrence rate 29.2% (80 out of 274), progression rate 7.3% (20 out of 274), high-grade tumors 50.4% (138 out of 274), and carcinoma in situ present 9.9% (27 out of 274). Statistical trend analysis demonstrated significant linear associations between increasing anemia severity and higher recurrence rates (*p* < 0.001), increased progression rates (*p* = 0.002), greater prevalence of high-grade tumors (*p* < 0.001), and increased carcinoma in situ presence (*p* = 0.012). Cox proportional hazards regression with hemoglobin categories using normal hemoglobin (HGB > 12.2 g/dL) as reference showed for recurrence that mild anemia (HGB 10 to 12.2) had HR 1.52 (95% CI 1.12 to 2.07, *p* = 0.007) and severe anemia (HGB < 10) had HR 2.31 (95% CI 1.61 to 3.31, *p* < 0.001). For progression, mild anemia had HR 1.68 (95% CI 0.95 to 2.97, *p* = 0.076) and severe anemia had HR 3.12 (95% CI 1.72 to 5.67, *p* < 0.001). A clear dose–response relationship exists between anemia severity and adverse oncological outcomes. Severe anemia (less than 10 g/dL) is particularly associated with aggressive tumor features and substantially elevated risk of both recurrence (2.3-fold) and progression (3.1-fold). This gradient effect supports the biological plausibility of hemoglobin as a prognostic biomarker and suggests that degree of anemia may inform risk stratification.

Kaplan–Meier analyses demonstrated differential recurrence-free survival patterns based on hematologic parameter stratification (Figure 1). Among all evaluated hematologic parameters, log-rank testing confirmed a statistically significant difference only for hemoglobin (*p* < 0.001), while other parameters (white blood cell count, lymphocyte count, platelet count) did not achieve statistical significance in survival curve comparisons. Using the optimal cutoff of 12.2 g/dL for hemoglobin, survival analysis revealed that lower hemoglobin levels were significantly associated with poorer recurrence-free survival (Figure 1A).

Receiver operating characteristic (ROC) curve analyses evaluated the diagnostic performance of hematologic parameters for outcome prediction (Table 3). For recurrence prediction, hemoglobin exhibited moderate discriminative ability among evaluated parameters (AUC 0.692), achieving 72.6% sensitivity and 64.2% specificity at the optimal cutoff of 12.2 g/dL. This AUC value indicates moderate discriminative performance and should be interpreted with appropriate caution. Lymphocyte count showed modest predictive capability (AUC 0.556), with optimal performance at 1.46 × 10^3^/µL. For progression prediction, hemoglobin again performed best (AUC 0.678), providing 68.3% sensitivity and 69.9% specificity at a cutoff of 11.8 g/dL. These cutoff values were derived from internal data in a single-center cohort and require external validation before clinical implementation.

Univariable and multivariable Cox proportional hazards regression identified independent predictors of recurrence (Table 4). In multivariable models incorporating all hematologic parameters and clinical covariates, hemoglobin concentration (HR 0.75, 95% CI 0.68–0.83, *p* < 0.001), age (HR 1.02, 95% CI 1.00–1.03, *p* = 0.023), and the presence of carcinoma in situ (HR 1.69, 95% CI 1.13–2.53, *p* = 0.011) emerged as significant independent predictors of recurrence. Interpretation of these findings reveals that each 1 g/dL increase in hemoglobin concentration is associated with a 25% reduction in recurrence risk (HR 0.75), indicating a protective effect of higher hemoglobin levels. Conversely, each additional year of age is associated with a 2% increase in recurrence risk (HR 1.02), while the presence of carcinoma in situ confers a 69% increased risk of recurrence (HR 1.69) compared with its absence. These findings indicate that hemoglobin exerts a protective effect against recurrence, whereas advancing age and carcinoma in situ represent adverse prognostic factors.

For disease progression, univariable analysis identified hemoglobin (HR 0.71, 95% CI 0.61–0.83, *p* < 0.001), neutrophil count (HR 0.90, 95% CI 0.83–0.98, *p* = 0.016), high tumor grade (HR 2.41, 95% CI 1.31–4.43, *p* = 0.005), and carcinoma in situ (HR 2.18, 95% CI 1.24–3.82, *p* = 0.007) as significant predictors. In multivariable Cox regression analysis for progression, hemoglobin (HR 0.73, 95% CI 0.62–0.86, *p* < 0.001), high tumor grade (HR 2.15, 95% CI 1.16–3.98, *p* = 0.015), and carcinoma in situ (HR 1.92, 95% CI 1.09–3.39, *p* = 0.024) remained independent prognostic factors (Table 5).

Among the 548 patients, 316 (57.7%) were classified as high-risk and received BCG therapy, 147 (26.8%) as intermediate-risk receiving variable treatments, and 85 (15.5%) as low-risk. In the high-risk BCG-treated subgroup, hemoglobin remained a significant independent predictor of recurrence (HR 0.72, 95% CI 0.63–0.83, *p* < 0.001) and progression (HR 0.69, 95% CI 0.56–0.85, *p* < 0.001). The prognostic value of hemoglobin was consistent across risk groups, although the effect size was most pronounced in the high-risk category. Stratification by BCG response status (BCG responders n = 203, 64.2%; BCG non-responders n = 113, 35.8%) revealed that preoperative hemoglobin levels were significantly lower in non-responders (11.4 ± 2.2 vs. 13.0 ± 2.3 g/dL, *p* < 0.001).

## 4. Discussion

This comprehensive analysis of 548 patients with NMIBC demonstrates the independent prognostic value of select hematologic parameters, with hemoglobin concentration emerging as the strongest predictor of both recurrence (HR 0.75, *p* < 0.001) and progression (HR 0.73, *p* < 0.001) alongside conventional clinicopathological factors. The observed recurrence rate of 37.0% and progression rate of 10.9% align closely with international literature, supporting the external validity of our findings across diverse populations [22,23,24].

The robust association between lower hemoglobin levels and increased risk of both recurrence (11.8 ± 2.3 vs. 13.2 ± 2.4 g/dL, *p* < 0.001) and progression (11.2 ± 2.1 vs. 12.7 ± 2.4 g/dL, *p* < 0.001) is clinically meaningful and biologically plausible. Cancer-associated anemia has been linked to adverse outcomes in multiple urological malignancies [13,24,25]. In bladder cancer specifically, preoperative hemoglobin has demonstrated independent prognostic value in patients undergoing radical cystectomy for muscle-invasive disease [24,25], but comprehensive data in NMIBC cohorts have been limited. The biological mechanisms underlying this association include impaired oxygen delivery to tissues, chronic inflammation, diminished immune competence, and adverse alterations within the tumor microenvironment that may facilitate progression [13].

A distinctive feature of our study is the comprehensive evaluation of individual complete blood count components rather than relying solely on derived inflammatory ratios. While the NLR has gained widespread attention as a prognostic marker in bladder cancer [17,18,19], our findings suggest that individual CBC parameters—particularly hemoglobin—may offer superior discriminative performance when evaluated independently. The NLR showed prognostic value in several NMIBC cohorts [17,20], particularly for predicting BCG response [21]. However, our multivariable models incorporating individual leukocyte subpopulations alongside hemoglobin did not identify inflammatory ratios as independent predictors after accounting for their constituent cellular components, suggesting potential collinearity effects.

While the relationship between hematologic parameters and cancer prognosis is well established, our study provides several novel contributions to the NMIBC literature. First, we present one of the largest cohorts systematically evaluating multiple individual CBC parameters with extended follow-up duration. Second, we demonstrate that hemoglobin maintains its prognostic value across EAU risk groups and specifically within BCG-treated high-risk patients, where its predictive utility is most clinically relevant. Third, our stratification analyses by intravesical treatment type and BCG response status provide actionable insights into the clinical contexts where hemoglobin assessment may be most valuable. Fourth, we provide empirically derived cutoff values optimized for the NMIBC population rather than applying general anemia thresholds.

The observation that preoperative hemoglobin levels were significantly lower in BCG non-responders (11.4 ± 2.2 vs. 13.0 ± 2.3 g/dL, *p* < 0.001) represents a clinically relevant finding. This suggests that baseline hematologic status may influence immunotherapeutic response, potentially through effects on immune competence and inflammatory tone. Integration of hemoglobin levels into BCG treatment algorithms may enhance patient selection and inform decisions regarding early cystectomy in high-risk cases with unfavorable baseline hematologic profiles.

While hemoglobin demonstrated statistically significant associations with both recurrence and progression, its discriminative performance (AUC 0.692 for recurrence, 0.678 for progression) falls within the moderate range and is substantially lower than would be required for standalone clinical decision-making. These AUC values should be interpreted with appropriate caution for several reasons. First, cutoff values were derived from internal data using ROC analysis with the Youden index method. This approach is prone to overfitting, particularly in single-center retrospective cohorts where population-specific characteristics may not generalize to external settings. The optimal cutoff of 12.2 g/dL identified in our cohort may perform differently when applied to independent populations with different demographic profiles, comorbidity patterns, or laboratory standardization protocols. Second, moderate discriminative performance (AUC 0.65 to 0.75 range) indicates that hemoglobin alone is insufficient for accurate individual-level prediction. While statistically significant at the population level, the clinical utility of this biomarker is limited when used in isolation. The observed sensitivity (72.6%) and specificity (64.2%) at the optimal cutoff would result in substantial misclassification rates if applied as a binary decision tool. Third, single-center derivation limits the generalizability of our findings. Geographic variation in hemoglobin reference ranges, differences in laboratory measurement techniques, and population-specific factors (altitude, genetic background, dietary patterns) may all influence the optimal cutoff value. Multi-center validation studies encompassing diverse populations are essential before these thresholds can be recommended for widespread clinical use. Given these limitations, we emphasize that hemoglobin should be viewed as a complementary prognostic factor to be integrated into comprehensive risk models rather than a standalone diagnostic tool. Its primary value lies in its universal availability, cost-effectiveness, and potential to modestly enhance existing risk stratification systems when combined with established clinicopathological parameters.

An unexpected and notable finding in our cohort was the inverse association between neutrophil counts and adverse outcomes: patients experiencing recurrence or progression demonstrated significantly lower neutrophil counts compared to those without events (recurrence: 5.3 ± 3.0 vs. 6.1 ± 3.4 × 10^3^/µL, *p* < 0.001; progression: 4.7 ± 2.0 vs. 5.8 ± 3.3 × 10^3^/µL, *p* = 0.014). This pattern appears paradoxical given the extensive literature documenting that elevated NLR, driven primarily by increased neutrophil counts, is associated with worse oncological outcomes in urothelial carcinoma. Several potential explanations warrant consideration. First, the NLR is a ratio of two components: neutrophil count (numerator) and lymphocyte count (denominator). In our cohort, the prognostic value of inflammatory indices may be primarily driven by alterations in the lymphocyte compartment and hemoglobin levels rather than neutrophil elevation. Patients with lower hemoglobin demonstrated both recurrence and progression, potentially reflecting a distinct inflammatory phenotype characterized by chronic disease, immune dysregulation, and lymphocyte expansion rather than acute neutrophilia. Second, patients with more aggressive tumor biology may experience relative bone marrow suppression affecting multiple cell lineages, including neutrophils. Cancer-associated anemia (reflected in lower hemoglobin) may coexist with relative neutropenia in patients with advanced disease burden, creating an apparent inverse relationship between neutrophil counts and outcomes. Third, patients who developed recurrence or progression may have received more intensive prior treatments (BCG therapy, chemotherapy) that suppressed myeloid cell production. Although we attempted to account for treatment protocols, residual confounding from treatment-related myelosuppression cannot be entirely excluded in a retrospective analysis spanning 17 years. Fourth, our study included hematologic parameters measured within 30 days prior to initial TURBT. Patients with more symptomatic disease (hematuria, infection) may have presented earlier and had reactive leukocytosis at presentation, potentially associating higher neutrophil counts with less aggressive disease. Conversely, patients with indolent, asymptomatic tumors (potentially more biologically aggressive) may have had baseline measurements reflecting chronic inflammatory states rather than acute responses. Fifth, our cohort exhibited unique characteristics (70% male, mean age 66 years, 52% hypertensive, 27% diabetic) that may influence inflammatory profiles. Chronic comorbidities could alter baseline neutrophil dynamics in ways not captured by multivariable adjustment. Sixth, the possibility of a chance finding cannot be excluded. While the *p*-values were statistically significant, the clinical magnitude of the difference was relatively modest (0.8 to 1.1 × 10^3^/µL difference), and multivariable models showed that neutrophil count did not remain independently significant after adjustment for hemoglobin and other factors (multivariate HR 0.96, *p* = 0.220). This unexpected finding underscores the complexity of inflammatory biomarkers in cancer prognosis and highlights potential limitations of applying findings from muscle-invasive bladder cancer to the NMIBC population. The distinct tumor biology, treatment paradigms, and natural history of NMIBC may produce inflammatory signatures that differ from those observed in advanced disease. Until this finding is replicated in independent cohorts, the prognostic significance of neutrophil counts in NMIBC remains uncertain, and clinical decision-making should not rely on this parameter in isolation.

Based on the hemoglobin stratification findings and prognostic associations demonstrated in this study, we propose the following clinical considerations for incorporating preoperative hemoglobin assessment into NMIBC management. First, patients with HGB < 12.2 g/dL, particularly those with severe anemia (<10 g/dL), should be considered for more intensive cystoscopic surveillance protocols. Severe anemia (<10 g/dL) combined with high-risk tumor features warrants early discussion of radical cystectomy as a treatment option, given the 3.1-fold increased progression risk. Second, all NMIBC patients presenting with HGB < 12 g/dL should undergo comprehensive anemia workup including iron studies, vitamin B12 and folate levels, renal function assessment, and evaluation for occult bleeding sources. Correction of reversible causes (iron deficiency, nutritional deficiencies) prior to intravesical therapy may optimize treatment outcomes, though this requires prospective validation. Third, lower preoperative hemoglobin was significantly associated with BCG non-response in our cohort (11.4 ± 2.2 vs. 13.0 ± 2.3 g/dL, *p* < 0.001). Patients with HGB < 10 g/dL receiving BCG therapy should undergo closer monitoring with lower threshold for early radical cystectomy if inadequate therapeutic response is observed. Fourth, preoperative hemoglobin < 12.2 g/dL should be incorporated into prognostic discussions and shared decision-making regarding treatment intensity, surveillance intervals, and consideration of early radical cystectomy in appropriate candidates. The dose–response relationship between anemia severity and adverse outcomes provides quantifiable risk estimates for patient counseling. Fifth, hemoglobin levels could be integrated into existing EAU or EORTC risk stratification tools as an additional readily available parameter, pending external validation. These recommendations are preliminary and based on retrospective single-center data. Prospective validation in multicenter cohorts is required before incorporation into formal clinical practice guidelines. Hemoglobin should complement, not replace, established risk stratification tools. Clinical decisions must integrate multiple factors including tumor characteristics (grade, stage, size, multiplicity, carcinoma in situ), patient preferences, comorbid conditions, performance status, and institutional expertise. The optimal approach to anemia correction in NMIBC patients (timing, methods, impact on outcomes) requires dedicated prospective investigation.

The absence of disease-specific mortality analysis represents a limitation of this study. Overall survival data were collected but disease-specific mortality was not systematically adjudicated due to the retrospective nature of data collection and inconsistent availability of detailed cause-of-death information across the 17-year study period. Future prospective studies should incorporate comprehensive mortality endpoints with systematic adjudication.

Several limitations warrant consideration in the interpretation of these findings. The single-center retrospective design may limit the generalizability of results, although our findings demonstrate consistency with established international benchmarks. The retrospective nature of data collection introduces potential selection bias and precludes standardization of treatment protocols across the study period. Additionally, the analysis was confined to preoperative hematologic parameters obtained within 30 days of surgery, which may not capture dynamic changes in these biomarkers over time or their temporal relationship with disease progression.

The study population was predominantly male (70.3%), reflecting the typical demographic distribution of bladder cancer but potentially limiting applicability to female patients. Furthermore, we did not account for potential confounding factors such as concurrent medications, inflammatory conditions, or nutritional status that could influence hematologic parameters. The moderate discriminative performance of individual biomarkers (AUC 0.692 for hemoglobin) suggests that these parameters may be most valuable when integrated into comprehensive risk models rather than used in isolation.

Future prospective multicenter studies with standardized protocols, longer follow-up periods, and serial biomarker assessments would strengthen the evidence base and facilitate validation of these findings across diverse populations and healthcare settings.

Critical limitations must be acknowledged: These cutoff values were derived from a single-center cohort and are susceptible to overfitting. External validation in independent, prospectively collected multicenter cohorts is essential before clinical implementation. The clinical utility of hemoglobin-based risk stratification will ultimately depend on demonstration of improved outcomes in prospective validation studies and decision-curve analyses confirming net clinical benefit.

## 5. Conclusions

This study demonstrates that preoperative hemoglobin concentration is an independent predictor of recurrence (HR 0.75, *p* < 0.001) and progression (HR 0.73, *p* < 0.001) in NMIBC. The optimal cutoff of 12.2 g/dL achieved 72.6% sensitivity and 64.2% specificity for recurrence prediction. However, the moderate discriminative performance (AUC 0.692) indicates that hemoglobin should supplement rather than replace established risk tools. As a universally available and cost-effective biomarker, hemoglobin may complement existing risk stratification systems and inform clinical decision-making, particularly regarding surveillance intensity and early intervention in high-risk patients. Future studies should investigate the biological mechanisms underlying the hemoglobin-prognosis relationship and evaluate integration with molecular biomarkers to further enhance risk stratification in NMIBC.

## Figures and Tables

**Figure 1 medicina-62-00051-f001:**
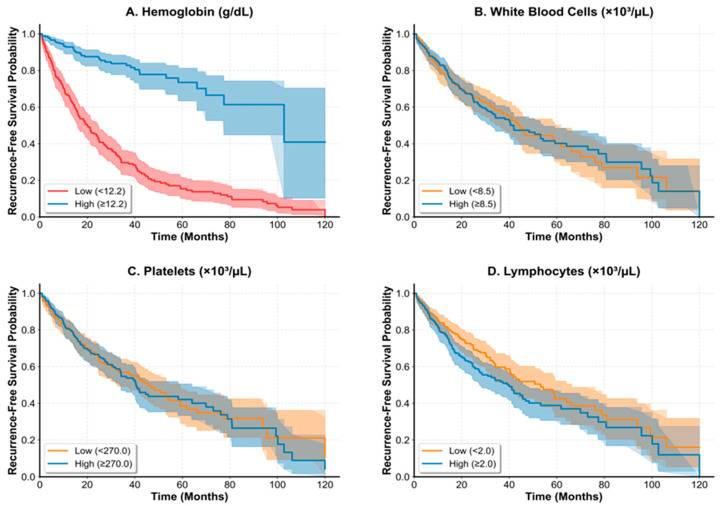
In all panels, solid lines represent patients with parameter values above the optimal cutoff (high group), and dashed lines represent patients with parameter values at or below the optimal cutoff (low group). Shaded areas indicate 95% confidence intervals. The *x*-axis shows follow-up time in months, and the *y*-axis shows recurrence-free survival probability. (**A**) Recurrence-free survival stratified by hemoglobin levels using the optimal cutoff of 12.2 g/dL. High hemoglobin (>12.2 g/dL, n = 274, solid line) versus low hemoglobin (≤12.2 g/dL, n = 274, dashed line). Log-rank test: *p* < 0.001. (**B**) Survival analysis according to white blood cell count levels using optimal cutoff of 3.50 × 10^3^/µL. Log-rank test: *p* = 0.423. (**C**) Survival curves based on platelet count stratification using optimal cutoff of 146 × 10^3^/µL. Log-rank test: *p* = 0.287. (**D**) Recurrence-free survival according to lymphocyte count levels using optimal cutoff of 1.46 × 10^3^/µL. Log-rank test: *p* = 0.156. Among all evaluated hematologic parameters, only hemoglobin (Panel (**A**)) demonstrated a statistically significant difference in recurrence-free survival between high and low groups.

**Table 1 medicina-62-00051-t001:** Sociodemographic and Baseline Patient Characteristics.

Characteristic	Value
**Sociodemographic Features**	
Age, years	66.3 ± 11.3
Male	385 (70.3%)
Female	163 (29.7%)
Body mass index, kg/m^2^	26.8 ± 4.2
**Comorbidities**	
Hypertension	285 (52.0%)
Diabetes mellitus	147 (26.8%)
Coronary artery disease	98 (17.9%)
Chronic obstructive pulmonary disease	76 (13.9%)
**Risk Factors**	
Smoking status—Never	203 (37.0%)
Smoking status—Former	198 (36.1%)
Smoking status—Current	147 (26.8%)
Occupational chemical exposure	89 (16.2%)
**Hematological Parameters**	
Hemoglobin, g/dL	12.5 ± 2.4
White blood cell count, ×10^3^/µL	8.8 ± 4.6
Lymphocyte count, ×10^3^/µL	2.2 ± 3.0
Monocyte count, ×10^3^/µL	0.7 ± 0.4
Neutrophil count, ×10^3^/µL	5.6 ± 3.2
Platelet count, ×10^3^/µL	271.6 ± 130.6
Mean platelet volume, fL	10.1 ± 1.1
**Tumor Characteristics**	
Tumor grade—High Grade	316 (57.7%)
Tumor grade—Low Grade	232 (42.3%)
Tumor stage—pT1	404 (73.7%)
Tumor stage—pTa	144 (26.3%)
Tumor number—Single	409 (74.6%)
Tumor number—Multiple	139 (25.4%)
Tumor size—<3 cm	305 (55.7%)
Tumor size—≥3 cm	243 (44.3%)
Carcinoma in situ—Absent	478 (87.2%)
Carcinoma in situ—Present	70 (12.8%)
Lymphovascular invasion—Absent	492 (89.8%)
Lymphovascular invasion—Present	56 (10.2%)
**Clinical Outcomes**	
Disease recurrence	203 (37.0%)
Disease progression	60 (10.9%)

Data are presented as mean ± standard deviation for normally distributed variables. For variables with non-normal distribution (lymphocyte count, platelet count), median with interquartile range (IQR) are also provided. Statistical tests were selected according to data distribution: independent *t*-test for normally distributed data (hemoglobin, white blood cell count, neutrophil count, monocyte count, mean platelet volume), Mann–Whitney U test for non-normally distributed data (lymphocyte count, platelet count).

**Table 2 medicina-62-00051-t002:** Comparative Analysis of Hematologic Parameters by Clinical Outcomes.

Disease Recurrence Analysis			
Parameter	No Recurrence (n = 345)	Recurrence (n = 203)	*p*-value
Hemoglobin, g/dL	13.2 ± 2.4	11.8 ± 2.3	<0.001
White blood cell count, ×10^3^/µL	9.0 ± 4.6	8.5 ± 4.8	0.168
Lymphocyte count, ×10^3^/µL ^†^	1.6 (1.1–2.2)	1.8 (1.2–2.5)	0.025
Monocyte count, ×10^3^/µL	0.7 ± 0.4	0.6 ± 0.3	0.122
Neutrophil count, ×10^3^/µL	6.1 ± 3.4	5.3 ± 3.0	<0.001
Platelet count, ×10^3^/µL ^†^	255 (205–330)	245 (195–315)	0.443
Mean platelet volume, fL	10.1 ± 1.1	10.0 ± 1.1	0.523
Disease Progression Analysis			
Parameter	No Progression (n = 488)	Progression (n = 60)	*p*-value
Hemoglobin, g/dL	12.7 ± 2.4	11.2 ± 2.1	<0.001
White blood cell count, ×10^3^/µL	9.0 ± 4.7	7.8 ± 3.0	0.065
Lymphocyte count, ×10^3^/µL ^†^	1.7 (1.2–2.3)	1.6 (1.1–2.0)	0.732
Monocyte count, ×10^3^/µL	0.7 ± 0.4	0.6 ± 0.2	0.245
Neutrophil count, ×10^3^/µL	5.8 ± 3.3	4.7 ± 2.0	0.014
Platelet count, ×10^3^/µL ^†^	252 (202–325)	240 (190–295)	0.189
Mean platelet volume, fL	10.1 ± 1.1	10.0 ± 1.2	0.512

Data are presented as mean ± standard deviation for normally distributed variables or as median (interquartile range) for non-normally distributed variables (marked with ^†^). Statistical tests were selected according to data distribution: independent *t*-test for normally distributed data, Mann–Whitney U test for non-normally distributed data (lymphocyte count, platelet count).

**Table 3 medicina-62-00051-t003:** ROC Analysis and Optimal Cutoff Values for Hematologic Parametersomparative Analysis of Hematologic Parameters by Clinical Outcomes.

Parameter	AUC	Cutoff Value	Sensitivity	Specificity	PPV	NPV	Accuracy
Disease Recurrence Prediction							
Hemoglobin, g/dL	0.692	12.2	0.726	0.642	0.567	0.789	0.676
White blood cell count, ×10^3^/µL	0.438	3.50	0.995	0.023	0.375	0.889	0.383
Lymphocyte count, ×10^3^/µL	0.556	1.46	0.798	0.316	0.407	0.727	0.495
Monocyte count, ×10^3^/µL	0.429	0.27	0.995	0.026	0.375	0.900	0.385
Neutrophil count, ×10^3^/µL	0.410	1.98	0.995	0.014	0.373	0.833	0.378
Platelet count, ×10^3^/µL	0.461	146.00	0.961	0.070	0.378	0.750	0.400
Mean platelet volume, fL	0.488	11.20	0.182	0.861	0.435	0.641	0.609
Disease Progression Prediction							
Hemoglobin, g/dL	0.678	11.8	0.683	0.699	0.219	0.933	0.697
White blood cell count, ×10^3^/µL	0.401	2.85	1.000	0.008	0.110	1.000	0.117
Lymphocyte count, ×10^3^/µL	0.530	2.14	0.517	0.623	0.144	0.913	0.611
Monocyte count, ×10^3^/µL	0.441	0.38	0.950	0.066	0.111	0.914	0.162
Neutrophil count, ×10^3^/µL	0.380	1.75	1.000	0.008	0.110	1.000	0.117
Platelet count, ×10^3^/µL	0.450	133.00	1.000	0.047	0.114	1.000	0.151
Mean platelet volume, fL	0.467	10.50	0.350	0.695	0.124	0.897	0.657

Optimal cutoff values were determined using the Youden index (sensitivity + specificity − 1). AUC: Area Under the Curve; PPV: Positive Predictive Value; NPV: Negative Predictive Value.

**Table 4 medicina-62-00051-t004:** Cox Regression Analysis for Disease Recurrence.

Variable	Total N	Events N	Univariate HR (95% CI)	*p*-Value	Multivariate HR (95% CI)	*p*-Value
Hemoglobin, g/dL	548	203	0.78 (0.71–0.86)	<0.001	0.75 (0.68–0.83)	<0.001
WBC count, ×10^3^/µL	548	203	0.98 (0.95–1.02)	0.342	1.02 (0.98–1.07)	0.298
Lymphocyte count, ×10^3^/µL	548	203	1.03 (0.98–1.08)	0.245	1.01 (0.96–1.06)	0.773
Monocyte count, ×10^3^/µL	548	203	0.82 (0.58–1.15)	0.248	0.72 (0.45–1.15)	0.169
Neutrophil count, ×10^3^/µL	548	203	0.92 (0.87–0.97)	0.003	0.96 (0.90–1.03)	0.220
Platelet count, ×10^3^/µL	548	203	1.00 (1.00–1.00)	0.684	1.00 (1.00–1.00)	0.742
Mean platelet volume, fL	548	203	0.96 (0.84–1.10)	0.562	1.04 (0.91–1.18)	0.562
Age, years	548	203	1.02 (1.00–1.03)	0.022	1.02 (1.00–1.03)	0.023
Gender (Female vs. Male)	548	203	1.08 (0.78–1.49)	0.645	1.12 (0.76–1.66)	0.564
High grade (vs. Low grade)	548	203	1.18 (0.89–1.56)	0.258	1.15 (0.87–1.52)	0.322
Multiple tumors (vs. Single)	548	203	0.94 (0.69–1.28)	0.697	0.88 (0.64–1.22)	0.442
Tumor size ≥3 cm (vs. <3 cm)	548	203	1.27 (0.96–1.69)	0.099	1.28 (0.94–1.75)	0.123
CIS present (vs. Absent)	548	203	1.58 (1.07–2.33)	0.020	1.69 (1.13–2.53)	0.011
LVI present (vs. Absent)	548	203	1.24 (0.70–2.18)	0.463	0.94 (0.51–1.74)	0.850

HR: Hazard Ratio; CI: Confidence Interval; CIS: Carcinoma in situ; LVI: Lymphovascular invasion.

**Table 5 medicina-62-00051-t005:** Cox Regression Analysis for Disease Recurrence.

Variable	Total N	Events N	Univariate HR (95% CI)	*p*-Value	Multivariate HR (95% CI)	*p*-Value
Hemoglobin, g/dL	548	60	0.71 (0.61–0.83)	<0.001	0.73 (0.62–0.86)	<0.001
WBC count, ×10^3^/µL	548	60	0.93 (0.86–1.01)	0.072	0.98 (0.90–1.07)	0.653
Lymphocyte count, ×10^3^/µL	548	60	0.98 (0.87–1.10)	0.734	0.97 (0.86–1.09)	0.614
Monocyte count, ×10^3^/µL	548	60	0.71 (0.38–1.31)	0.273	0.68 (0.35–1.33)	0.259
Neutrophil count, ×10^3^/µL	548	60	0.90 (0.83–0.98)	0.016	0.94 (0.86–1.03)	0.187
Platelet count, ×10^3^/µL	548	60	1.00 (0.99–1.00)	0.197	1.00 (1.00–1.00)	0.431
Mean platelet volume, fL	548	60	0.93 (0.74–1.17)	0.526	0.98 (0.77–1.24)	0.856
Age, years	548	60	1.01 (0.99–1.04)	0.243	1.01 (0.98–1.03)	0.614
Gender (Female vs. Male)	548	60	1.14 (0.65–2.00)	0.648	1.21 (0.66–2.22)	0.542
High grade (vs. Low grade)	548	60	2.41 (1.31–4.43)	0.005	2.15 (1.16–3.98)	0.015
Multiple tumors (vs. Single)	548	60	1.23 (0.72–2.10)	0.446	1.14 (0.66–1.98)	0.637
Tumor size ≥3 cm (vs. <3 cm)	548	60	1.46 (0.89–2.39)	0.132	1.38 (0.84–2.28)	0.207
CIS present (vs. Absent)	548	60	2.18 (1.24–3.82)	0.007	1.92 (1.09–3.39)	0.024
LVI present (vs. Absent)	548	60	1.58 (0.78–3.21)	0.203	1.34 (0.65–2.76)	0.427

HR: Hazard Ratio; CI: Confidence Interval; CIS: Carcinoma in situ; LVI: Lymphovascular invasion.

## Data Availability

The datasets generated and analyzed during the current study are available from the corresponding author upon reasonable request, subject to institutional privacy regulations.

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
