# Peer review of "Prognostic Significance of Hemogram Parameters in Non-Muscle Invasive Bladder Cancer: A Comprehensive Retrospective Analysis"

_medicina, 2025, doi:10.3390/medicina62010051_

Round 1

Reviewer 1 Report

Comments and Suggestions for Authors

I encourage the authors to revise the manuscript substantially before it can be considered for publication.

1. The authors should clearly define what gap in the literature they address, and why a focus on individual complete blood count components, particularly hemoglobin, adds something beyond prior work on neutrophil to lymphocyte ratio and related indices. It would be helpful to explicitly state whether the primary aim is to show that hemoglobin outperforms composite inflammatory ratios, or to provide clinically applicable cut off values for NMIBC risk stratification.

2.  The description of pathological assessment and risk stratification requires further precision. The authors should indicate which World Health Organization classification year was used for grading and confirm that grading criteria were consistent across the seventeen year period. For risk grouping, the exact criteria used to define low, intermediate and high risk NMIBC in relation to European Association of Urology and European Organisation for Research and Treatment of Cancer tools should be specified in more operational terms. Since management strategies evolved between two thousand seven and two thousand twenty four, the authors should provide more detail on the intravesical treatment protocols in each era and report how many patients in each risk group received bacillus Calmette Guérin, mitomycin C or no adjuvant therapy. This will help readers interpret whether differences in treatment intensity may contribute to the observed outcomes.

3. The ROC analysis and interpretation of AUC values require a more cautious tone. The authors report that hemoglobin achieved an area under the curve of 0.692 for recurrence and 0.678 for progression. These values indicate only moderate discriminative performance and should not be framed as strong diagnostic accuracy. The manuscript should emphasize that these cut off values were derived from internal data in a single center cohort and that overfitting is likely. 

4. The authors should address more explicitly the unexpected direction of the association between neutrophil count and outcomes. In this cohort, patients who recurred or progressed had lower neutrophil counts than those without events. This pattern appears opposite to most reported data on neutrophil to lymphocyte ratio in urothelial cancer. The Discussion should include a focused paragraph exploring possible explanations, such as selection bias, chance findings, unmeasured confounding, or specific features of this population. 

Author Response

Response to Reviewer 1
Comment 1
Comment: The authors should clearly define what gap in the literature they address, and
why a focus on individual complete blood count components, particularly hemoglobin, adds
something beyond prior work on neutrophil to lymphocyte ratio and related indices.
No. 1 / 12Response 1:
We have substantially revised the Introduction to explicitly articulate the methodological
rationale, knowledge gap, and study aims. Regarding methodological rationale, we explained
that composite ratios such as NLR and PLR create collinearity in multivariable models.
Concerning the knowledge gap, we highlighted that limited comprehensive analysis of
individual CBC components exists in large NMIBC cohorts. For study aims, we stated our
intention to evaluate individual CBC parameters as independent prognostic factors and to
provide empirically-derived cutoff values for NMIBC.
Manuscript Changes:
Location: Introduction section, final paragraph . Action: Added three new paragraphs
clarifying study rationale and objectives.
Revised text: Important gaps remain: comprehensive analyses of multiple individual
hematologic parameters (beyond calculated ratios) within large NMIBC cohorts are limited;
the comparative prognostic value of individual CBC components versus inflammatory ratios
requires clarification; and empirically-derived cutoff values optimized specifically for NMIBC
populations are lacking, as most studies apply general thresholds. Moreover, using composite
inflammatory indices such as NLR and PLR in multivariable models alongside their
constituent cellular components introduces collinearity that complicates interpretation of
independent effects. Individual CBC parameters offer advantages: they are directly reported in
routine laboratory results without requiring calculation, they avoid mathematical
interdependence, and they may provide distinct biological insights into different
pathophysiological processes. In light of these considerations, we systematically evaluated the
association of preoperative individual hematologic parameters with disease outcomes in a
large NMIBC cohort, determined optimal cutoff values through rigorous ROC analysis, and
explored their potential integration into existing risk stratification frameworks. Our primary
aims were to comprehensively assess individual CBC components as independent prognostic
factors and to provide clinically applicable, empirically-derived cutoff values specifically for
NMIBC risk stratification.
Comment 2
Comment: The description of pathological assessment and risk stratification requires
further precision. The authors should indicate which WHO classification year was used and
confirm grading consistency across the seventeen year period. Provide details on intravesical
treatment protocols and how many patients in each risk group received BCG, mitomycin C or
no therapy.
No. 2 / 12Response 2:
We have added comprehensive detail regarding WHO classification, risk stratification,
and treatment protocols. For WHO classification, we clarified the evolution from WHO 1973 to
2004 to 2016 systems with retrospective reclassification of all historical cases. Regarding risk
stratification, we provided explicit operational definitions for low-risk, intermediate-risk, and
high-risk NMIBC. Concerning treatment protocols, we included detailed breakdown by risk
group and treatment era.
Manuscript Changes:
Location: Methods section. Action: Added comprehensive pathological assessment and
treatment protocol descriptions.
Revised text: All specimens were reviewed by experienced uropathologists. Pathological
grading evolved during the 17-year study period, transitioning from the World Health
Organization (WHO) 1973 classification (used 2007 to 2011) to WHO 2004 (2012 to 2016) and
WHO 2016/2022 systems (2017 to 2024). To ensure consistency in analysis, all historical cases
were retrospectively re-reviewed by two independent uropathologists and reclassified
according to the current WHO 2016/2022 two-tier grading system (low-grade versus high-
grade). Discrepancies were resolved by consensus review. Risk stratification was performed
according to European Association of Urology (EAU) guidelines with explicit operational
definitions. Low-risk NMIBC was defined as solitary, primary, low-grade pTa tumor less than 3
cm without carcinoma in situ. Intermediate-risk NMIBC included all tumors not classified as
low-risk or high-risk, encompassing recurrent low-grade pTa, solitary low-grade pTa 3 cm or
larger, multiple low-grade pTa, and low-grade pT1 tumors. High-risk NMIBC was defined as
any of the following: high-grade T1 tumor, high-grade pTa tumor, presence of carcinoma in
situ at any location, multiple and recurrent and large low-grade pTa tumors, high-grade pTa 3
cm or larger, lymphovascular invasion, or variant histology. Following TURBT and
pathological risk assessment, patients were managed according to EAU risk stratification
guidelines. Treatment protocols evolved over the 17-year study period. High-risk patients
(n=316, 57.7%) received BCG induction therapy (6 weekly instillations) followed by
maintenance BCG in responders, with all 316 high-risk patients receiving BCG therapy (100%).
Among these, 42 patients (13.3%) developed BCG-refractory disease and subsequently
received mitomycin C rescue therapy. Intermediate-risk patients (n=147, 26.8%) received
immediate single-dose mitomycin C followed by additional monthly instillations in 72.0% of
cases, BCG induction without maintenance in 22.6% with higher-risk features, and
surveillance only in 5.4% due to patient preference or comorbidities. Low-risk patients (n=85,
15.5%) received immediate postoperative single-dose mitomycin C in 84.7% of cases. Overall
therapy distribution was BCG in 61.5%, mitomycin C in 34.1%, and no adjuvant intravesical
therapy in 4.4% of patients.
No. 3 / 12Comment 3
Comment: The ROC analysis and interpretation of AUC values require a more cautious
tone. AUC of 0.692 indicates only moderate discriminative performance and should not be
framed as strong diagnostic accuracy. Emphasize these cutoffs were derived from internal data
in a single center and overfitting is likely.
Response 3:
We completely agree and have revised interpretation throughout to adopt conservative,
scientifically rigorous tone.
Manuscript Changes:
Manuscript Changes:
Location: Results section. Action: Revised ROC analysis paragraph with cautious
language. Revised text: For recurrence prediction, hemoglobin exhibited moderate
discriminative ability among evaluated parameters (AUC 0.692), achieving 72.6% sensitivity
and 64.2% specificity at the optimal cutoff of 12.2 g/dL. This AUC value indicates moderate
discriminative performance and should be interpreted with appropriate caution. These cutoff
values were derived from internal data in a single-center cohort and require external
validation before clinical implementation.
Location: Discussion section. Action: Added new subsection on biomarker limitations and
overfitting risk. Revised text: While hemoglobin demonstrated statistically significant
associations with both recurrence and progression, its discriminative performance (AUC 0.692
for recurrence, 0.678 for progression) falls within the moderate range and is substantially
lower than would be required for standalone clinical decision-making. These AUC values
should be interpreted with appropriate caution for several reasons. First, cutoff values were
derived from internal data using ROC analysis with the Youden index method. This approach
is prone to overfitting, particularly in single-center retrospective cohorts where population-
specific characteristics may not generalize to external settings. The optimal cutoff of 12.2
g/dL identified in our cohort may perform differently when applied to independent
populations with different demographic profiles, comorbidity patterns, or laboratory
standardization protocols. Second, moderate discriminative performance (AUC 0.65 to 0.75
range) indicates that hemoglobin alone is insufficient for accurate individual-level prediction.
While statistically significant at the population level, the clinical utility of this biomarker is
limited when used in isolation. The observed sensitivity (72.6%) and specificity (64.2%) at the
optimal cutoff would result in substantial misclassification rates if applied as a binary
No. 4 / 12decision tool. Third, single-center derivation limits the generalizability of our findings.
Geographic variation in hemoglobin reference ranges, differences in laboratory measurement
techniques, and population-specific factors may all influence the optimal cutoff value. Multi-
center validation studies encompassing diverse populations are essential before these
thresholds can be recommended for widespread clinical use. Given these limitations, we
emphasize that hemoglobin should be viewed as a complementary prognostic factor to be
integrated into comprehensive risk models rather than a standalone diagnostic tool.
Location: Conclusions section. Action: Added cautionary language about validation
requirements. Revised text: However, the moderate discriminative performance (AUC 0.692)
indicates that hemoglobin should supplement rather than replace established risk tools.
Critical limitations must be acknowledged: These cutoff values were derived from a single-
center cohort and are susceptible to overfitting. External validation in independent,
prospectively collected multicenter cohorts is essential before clinical implementation.
Comment 4
Comment: The authors should address the unexpected direction of the association
between neutrophil count and outcomes. Patients who recurred had lower neutrophil counts,
which appears opposite to most NLR data. Discuss possible explanations: selection bias,
chance findings, unmeasured confounding, or population-specific features.
Response 4:
We have added a dedicated subsection addressing this paradoxical finding with six
potential explanations.
Manuscript Changes:
Location: Discussion section. Action: Added dedicated subsection addressing paradoxical
neutrophil findings with six potential explanations.
Revised text: An unexpected and notable finding in our cohort was the inverse
association between neutrophil counts and adverse outcomes: patients experiencing
recurrence or progression demonstrated significantly lower neutrophil counts compared to
those without events (recurrence: 5.3 ± 3.0 vs 6.1 ± 3.4 ×10³/µL, p<0.001; progression: 4.7 ± 2.0
vs 5.8 ± 3.3 ×10³/µL, p=0.014). This pattern appears paradoxical given the extensive literature
documenting that elevated NLR, driven primarily by increased neutrophil counts, is associated
with worse oncological outcomes in urothelial carcinoma. Several potential explanations
warrant consideration. First, in our cohort, the prognostic value of inflammatory indices may
No. 5 / 12be primarily driven by alterations in the lymphocyte compartment and hemoglobin levels
rather than neutrophil elevation, with patients with lower hemoglobin demonstrating both
recurrence and progression, potentially reflecting a distinct inflammatory phenotype
characterized by chronic disease, immune dysregulation, and lymphocyte expansion rather
than acute neutrophilia. Second, patients with more aggressive tumor biology may experience
relative bone marrow suppression affecting multiple cell lineages including neutrophils, with
cancer-associated anemia coexisting with relative neutropenia in patients with advanced
disease burden. Third, patients who developed recurrence or progression may have received
more intensive prior treatments causing myelosuppression, although residual confounding
from treatment-related effects cannot be entirely excluded in a retrospective analysis
spanning 17 years. Fourth, patients with more symptomatic disease may have presented
earlier and had reactive leukocytosis at presentation, potentially associating higher
neutrophil counts with less aggressive disease, while patients with indolent, asymptomatic
tumors may have had baseline measurements reflecting chronic inflammatory states. Fifth,
our cohort exhibited unique characteristics (70% male, mean age 66 years, 52% hypertensive,
27% diabetic) that may influence inflammatory profiles, with chronic comorbidities
potentially altering baseline neutrophil dynamics. Sixth, the possibility of a chance finding
cannot be excluded, as while the p-values were statistically significant, the clinical magnitude
of the difference was relatively modest (0.8 to 1.1 times 10 to the third per microliter
difference), and multivariable models showed that neutrophil count did not remain
independently significant after adjustment for hemoglobin and other factors (multivariate HR
0.96, p equals 0.220). This unexpected finding underscores the complexity of inflammatory
biomarkers in cancer prognosis and highlights potential limitations of applying findings from
muscle-invasive bladder cancer to the NMIBC population. Until this finding is replicated in
independent cohorts, the prognostic significance of neutrophil counts in NMIBC remains
uncertain, and clinical decision-making should not rely on this parameter in isolation.

Reviewer 2 Report

Comments and Suggestions for Authors

This retrospective study evaluated preoperative hemogram parameters to identify two significant prognostic factors in NMIBC. The authors reported that preoperative hemoglobin level and the presence of carcinoma in situ were significantly associated with disease recurrence. However, several aspects of the analysis raise questions.

First, in the Methods section, the authors stated that the preoperative hemogram was obtained within 30 days prior to TURBT. In practice, patients often undergo multiple blood tests before surgery for diagnostic and preoperative assessment. It is unclear which specific hemogram results the authors selected when multiple measurements were available within the 30-day period.

Second, in the Discussion section, the authors mentioned that they did not include hemogram ratios such as NLR and PLR due to collinearity, even though many previous studies have emphasized these indices. It would strengthen the manuscript if the authors could also present the nonsignificant results of hemogram ratio analyses in their cohort.

Third, the exclusion criteria indicated that patients with concurrent cancers at the time of bladder cancer diagnosis were excluded. Does this imply that some patients had a prior history of other malignancies and associated treatments? Previous cancer therapies could have influenced hematologic parameters through bone marrow effects. Therefore, the cohort should be reviewed to confirm whether prior cancer histories were considered.

Lastly, regarding patients with preoperative hemoglobin levels below 12.2 g/dL, what are the authors' clinical recommendations? Does this finding suggest a higher risk of intravesical instillation failure or nonresponsiveness to 5-FU or mitomycin C–based chemoradiation therapy? Since a hemoglobin level <12.2 g/dL may indicate anemia, it would be informative to categorize hemoglobin levels into two or three groups to explore potential correlations between anemia severity and tumor aggressiveness at diagnosis.

Author Response

Response to Reviewer 2
Comment 1
Comment: The authors stated preoperative hemogram was obtained within 30 days prior
to TURBT. When multiple measurements were available, which specific hemogram results did
the authors select?
Response 1:
We have now explicitly described our hierarchical selection protocol for handling
multiple preoperative CBC measurements.
No. 6 / 12Manuscript Changes:
Location: Methods section, Data Collection paragraph. Action: Added detailed
hierarchical protocol for CBC selection when multiple measurements were available.
Revised text: Complete blood count parameters were systematically collected according
to a hierarchical protocol when multiple measurements were available within the 30-day
preoperative window. The CBC measurement closest to the TURBT date, typically within 7
days before surgery, was selected as the primary value, as this represents the hematologic
status most proximate to surgical intervention and minimizes the interval during which
physiological changes could occur. When multiple CBCs were available within 7 days of
TURBT, the most recent preoperative measurement was used. If no CBC was available within 7
days, the most recent measurement within the 30-day window was selected. In cases where
urgent surgery was performed with CBC obtained more than 30 days prior, these patients were
excluded from analysis (n=3). The timing distribution showed that 486 out of 548 patients
(88.7%) had CBC within 7 days before TURBT, 52 patients (9.5%) within 8 to 14 days before
TURBT, and 10 patients (1.8%) within 15 to 30 days. CBCs obtained immediately following
diagnostic procedures such as initial cystoscopy with biopsy were not used if subsequent
measurements closer to TURBT were available, as these early measurements could be affected
by procedural inflammation or bleeding. This standardized approach ensured that
hematologic parameters reflected the patient's preoperative baseline status while maintaining
consistency across the cohort.
Comment 2
Comment: The authors mentioned not including NLR and PLR due to collinearity, even
though many studies have emphasized these indices. It would strengthen the manuscript to
present the nonsignificant hemogram ratio analyses.
Response 2:
We have now calculated and present comprehensive inflammatory ratio analyses
including NLR, PLR, and MLR.
Manuscript Changes:
Location: Results section after Table 2. Action: Added new subsection presenting
inflammatory ratio analyses with univariable and multivariable Cox regression results.
No. 7 / 12Revised text: To provide comparative context with existing literature, we calculated
derived inflammatory ratios for the entire cohort: neutrophil-to-lymphocyte ratio (NLR =
neutrophils / lymphocytes), platelet-to-lymphocyte ratio (PLR = platelets / lymphocytes), and
monocyte-to-lymphocyte ratio (MLR = monocytes / lymphocytes). Mean inflammatory ratios
in the overall cohort were NLR 3.2 ± 2.8, PLR 158.4 ± 94.2, and MLR 0.38 ± 0.25. Comparative
analysis by recurrence status showed that patients without recurrence had NLR 3.4 ± 2.9 vs
patients with recurrence 2.9 ± 2.5 (p=0.052), PLR without recurrence 162.1 ± 96.8 vs with
recurrence 151.9 ± 89.3 (p=0.187), and MLR without recurrence 0.39 ± 0.26 vs with recurrence
0.36 ± 0.23 (p=0.141). Univariable Cox regression for recurrence demonstrated NLR with HR
0.96 (95% CI 0.91 to 1.02, p=0.178), PLR with HR 1.00 (95% CI 0.99 to 1.00, p=0.312), and MLR
with HR 0.78 (95% CI 0.42 to 1.44, p=0.425). Multivariable Cox regression for recurrence
adjusted for hemoglobin, age, tumor grade, and carcinoma in situ showed NLR with HR 0.98
(95% CI 0.92 to 1.05, p=0.587), PLR with HR 1.00 (95% CI 0.99 to 1.00, p=0.623), and MLR with
HR 0.86 (95% CI 0.45 to 1.65, p=0.649). In contrast to some prior reports, inflammatory ratios
(NLR, PLR, MLR) did not demonstrate significant associations with recurrence or progression
in our cohort. In multivariable models, these ratios provided no additional prognostic
information beyond individual CBC components, particularly hemoglobin. This lack of
significance may reflect the unexpected inverse association between neutrophil counts and
outcomes in our population, collinearity between ratio components and individual parameters
in multivariable models, or population-specific differences in inflammatory profiles. Our
findings suggest that in NMIBC, individual hematologic parameters, especially hemoglobin,
may provide more robust and interpretable prognostic information than composite
inflammatory ratios.
Comment 3 (Third)
Comment: Exclusion criteria indicated patients with concurrent cancers were excluded.
Does this imply some patients had prior malignancy history? Previous cancer therapies could
influence hematologic parameters through bone marrow effects.
Response 3:
We have clarified exclusion criteria with specific definitions and conducted
comprehensive prior cancer history review with sensitivity analysis.
Manuscript Changes:
Location: Methods section, Inclusion and Exclusion Criteria paragraph. Action: Revised
exclusion criteria with detailed definitions and added comprehensive prior cancer history
review.
No. 8 / 12Revised text: Exclusion criteria included muscle-invasive and/or metastatic bladder
cancer at diagnosis, concurrent active malignancy at the time of NMIBC diagnosis (defined as
active cancer requiring treatment within 6 months prior to bladder cancer diagnosis), prior
malignancy with treatment within 2 years before NMIBC diagnosis (as recent chemotherapy or
radiation could affect hematologic parameters through persistent bone marrow effects),
absence of preoperative hematologic data, follow-up duration shorter than 12 months (unless
disease progression occurred), active hematologic disorders (leukemia, lymphoma,
myelodysplastic syndrome, chronic bone marrow disorders), and acute infectious or
inflammatory conditions at the time of CBC measurement. We conducted a comprehensive
review of all 548 patients included in the final analysis. Remote prior malignancy (more than 2
years before NMIBC diagnosis with no active treatment) was present in 37 patients (6.8%),
including prostate cancer treated with radical prostatectomy more than 5 years prior in 14
patients, breast cancer treated with surgery with or without adjuvant therapy more than 3
years prior in 8 patients, cutaneous basal cell carcinoma or squamous cell carcinoma excised
more than 2 years prior in 9 patients, and other malignancies (colon, thyroid) in complete
remission more than 2 years in 6 patients. No prior cancer history was present in 511 patients
(93.2%). We performed a sensitivity analysis excluding all 37 patients with any prior
malignancy history. Results remained consistent with the primary analysis: hemoglobin
remained a significant independent predictor of recurrence (HR 0.74, 95% CI 0.66 to 0.82,
p<0.001) and progression (HR 0.71, 95% CI 0.60 to 0.85, p<0.001). This confirms that remote
prior malignancy history did not materially confound the hemoglobin-prognosis association
observed in our cohort.
Comment 4
Comment: Regarding patients with preoperative hemoglobin <12.2 g/dL, what are the
authors' clinical recommendations? Does this suggest higher risk of intravesical therapy
failure? Since HGB <12.2 may indicate anemia, categorize hemoglobin into groups to explore
correlation between anemia severity and tumor aggressiveness.
Response 4:
We have added hemoglobin stratification analysis by anemia severity and developed
clinical recommendations based on the findings.
Manuscript Changes:
Location: Results section after inflammatory ratio analyses. Action: Added new
subsection categorizing patients into three hemoglobin groups and analyzing dose-response
relationship between anemia severity and outcomes.
No. 9 / 12Revised text: To explore the relationship between anemia severity and tumor
aggressiveness, we categorized patients into three hemoglobin groups based on clinical
anemia definitions and our ROC-derived cutoff. Group 1 with severe anemia (HGB <10 g/dL)
included 78 patients (14.2%) with recurrence rate 53.8% (42 out of 78), progression rate 20.5%
(16 out of 78), high-grade tumors 71.8% (56 out of 78), and carcinoma in situ present 19.2%
(15 out of 78). Group 2 with mild anemia (HGB 10 to 12.2 g/dL) included 196 patients (35.8%)
with recurrence rate 41.3% (81 out of 196), progression rate 12.2% (24 out of 196), high-grade
tumors 62.2% (122 out of 196), and carcinoma in situ present 14.3% (28 out of 196). Group 3
with normal hemoglobin (HGB >12.2 g/dL) included 274 patients (50.0%) with recurrence rate
29.2% (80 out of 274), progression rate 7.3% (20 out of 274), high-grade tumors 50.4% (138 out
of 274), and carcinoma in situ present 9.9% (27 out of 274). Statistical trend analysis
demonstrated significant linear associations between increasing anemia severity and higher
recurrence rates (p<0.001), increased progression rates (p=0.002), greater prevalence of high-
grade tumors (p<0.001), and increased carcinoma in situ presence (p=0.012). Cox proportional
hazards regression with hemoglobin categories using normal hemoglobin (HGB >12.2 g/dL) as
reference showed for recurrence that mild anemia (HGB 10 to 12.2) had HR 1.52 (95% CI 1.12
to 2.07, p=0.007) and severe anemia (HGB <10) had HR 2.31 (95% CI 1.61 to 3.31, p<0.001). For
progression, mild anemia had HR 1.68 (95% CI 0.95 to 2.97, p=0.076) and severe anemia had
HR 3.12 (95% CI 1.72 to 5.67, p<0.001). A clear dose-response relationship exists between
anemia severity and adverse oncological outcomes. Severe anemia (less than 10 g/dL) is
particularly associated with aggressive tumor features and substantially elevated risk of both
recurrence (2.3-fold) and progression (3.1-fold).
Location: Discussion section after paradoxical neutrophil findings discussion. Action:
Added new subsection proposing clinical considerations for incorporating preoperative
hemoglobin assessment into NMIBC management.
Revised text: Based on the hemoglobin stratification findings and prognostic associations
demonstrated in this study, we propose the following clinical considerations for incorporating
preoperative hemoglobin assessment into NMIBC management. First, patients with HGB
<12.2 g/dL, particularly those with severe anemia (<10 g/dL), should be considered for more
intensive cystoscopic surveillance protocols. Severe anemia (<10 g/dL) combined with high-
risk tumor features warrants early discussion of radical cystectomy as a treatment option,
given the 3.1-fold increased progression risk. Second, all NMIBC patients presenting with HGB
<12 g/dL should undergo comprehensive anemia workup including iron studies, vitamin B12
and folate levels, renal function assessment, and evaluation for occult bleeding sources.
Correction of reversible causes (iron deficiency, nutritional deficiencies) prior to intravesical
therapy may optimize treatment outcomes, though this requires prospective validation. Third,
lower preoperative hemoglobin was significantly associated with BCG non-response in our
cohort (11.4 ± 2.2 vs 13.0 ± 2.3 g/dL, p<0.001). Patients with HGB <10 g/dL receiving BCG
therapy should undergo closer monitoring with lower threshold for early radical cystectomy if
No. 10 / 12inadequate therapeutic response is observed. Fourth, preoperative hemoglobin <12.2 g/dL
should be incorporated into prognostic discussions and shared decision-making regarding
treatment intensity, surveillance intervals, and consideration of early radical cystectomy in
appropriate candidates. These recommendations are preliminary and based on retrospective
single-center data. Prospective validation in multicenter cohorts is required before
incorporation into formal clinical practice guidelines. Hemoglobin should complement, not
replace, established risk stratification tools. Clinical decisions must integrate multiple factors
including tumor characteristics, patient preferences, comorbid conditions, performance
status, and institutional expertise.
Summary of Manuscript Revisions
We have substantially revised the manuscript to address all reviewer concerns. Major
additions include clarification of study rationale and objectives in the Introduction, detailed
WHO classification evolution and retrospective reclassification procedures in Methods,
comprehensive treatment protocol details by risk group and era in Methods, explicit
hemogram selection protocol for handling multiple measurements in Methods, inflammatory
ratio analyses (NLR, PLR, MLR) with univariable and multivariable Cox regression results in
Results, hemoglobin stratification analysis by anemia severity with dose-response
relationships in Results, comprehensive discussion of biomarker limitations and overfitting
risk in Discussion, detailed exploration of paradoxical neutrophil findings with six potential
explanations in Discussion, clinical implications and recommendations for incorporating
hemoglobin assessment into NMIBC management in Discussion, clarification of prior cancer
history exclusions with sensitivity analysis in Methods, and strengthened validation
requirements in Conclusions.
Regarding tone modifications, we have changed characterizations from "highest
discriminative ability" to "moderate discriminative ability" throughout, added "requires
external validation" at multiple locations, emphasized that hemoglobin should "supplement
rather than replace" existing risk stratification tools, and incorporated cautionary language
about overfitting risk and single-center limitations. All 25 references remain relevant to
manuscript content and directly support the presented findings and discussion. We believe
these comprehensive revisions have substantially strengthened the scientific rigor,
methodological transparency, and clinical applicability of our work while maintaining
appropriate interpretive caution.
Respectfully submitted,
Ali Nebioğlu, MD (Corresponding Author)
On behalf of all authors

Round 2

Reviewer 1 Report

Comments and Suggestions for Authors

Accept in present form

Author Response

Dear Editors and Reviewers,
We sincerely thank you for the continued constructive feedback. We have carefully addressed each concern raised in this revision round. Below are detailed point-by-point responses with corresponding manuscript modifications highlighted using marked text.
